# *"When we have served meat, my husband comes first"*: A qualitative analysis of child nutrition among urban and rural communities of Rwanda

**Maria Qambayot Albin**[1‡], **Gloria Igihozo**[1,2‡], **Shuko Musemangezhi**[2‡], **Edith Nachizya Namukanga**[3], **Theogene Uwizeyimana**[2], **Gebremariam Alemayehu**[4], **Abebe Bekele**[5], **Rex Wong**[2,6], **Chester Kalinda**[2,7] *

1 Centre for One Health, University of Global Health Equity (UGHE), Kigali, Rwanda, 2 Bill and Joyce Cummings Institute of Global Health, University of Global Health Equity (UGHE), Kigali, Rwanda, 3 School of Health Sciences, UNICAF University, Lusaka, Zambia, 4 Catholic Relief Services, Rwanda Country Program, Kigali, Rwanda, 5 School of Medicine, University of Global Health Equity (UGHE), Kigali, Rwanda, 6 School of Public Health, Yale University, New Haven, Connecticut, United States of America, 7 School of Nursing and Public Health (SNPH), Discipline of Public Health Medicine, Howard College Campus, University of KwaZulu-Natal, Durban, South Africa

‡ MQA, GI and SM are contributed equally and share first authorship
* ckalinda@gmail.com, ckalinda@ughe.org

**Data Availability Statement:** The dataset was uploaded to the Mendeley data repository. (DOI: 10.17632/r9xkh9wtxn.1).

## Abstract

### Background

Stunting among children under five years of age is a global public health concern, especially in low-and middle-income settings. Emerging evidence suggests a gradual reduction in the overall prevalence of stunting in Rwanda, necessitating a qualitative understanding of the contributing drivers to help develop targeted and effective strategies. This qualitative study explored the lived experiences of women and men to identify key issues that influence childhood nutrition and stunting as well as possible solutions to address the problem.

### Methods

Ten (10) focus group discussions (FGDs) were conducted with fathers and mothers of children under five years of age from five districts, supplemented by forty (40) in-depth interviews (IDIs) with Nurses and Community Health Workers (CHWs). Transcripts were coded inductively and analysed thematically using Dedoose (version 9.0.86).

### Results

Three themes emerged: (1) Awareness of a healthy diet for pregnant women, infants, and children with subthemes Knowledge about maternal and child nutrition and feeding practices; (2) Personal and food hygiene is crucial while handling, preparing, and eating food with subthemes, food preparation practices and the feeding environment (3) factors influencing healthy eating among pregnant women, infants, and children with subthemes; Barriers and facilitators to healthy eating among pregnant women and children.

**Funding:** USAID Rwanda through the Gikuriro Kuri Bose - Inclusive Nutrition and Early Childhood Development (INECD).

**Competing interests:** The authors have declared that no competing interests exist.

## Conclusion

Several factors influence child stunting, and strategies to address them should recognise the cultural and social contexts of the problem. Prioritisation of nutrition-based strategies is vital and should be done using a multifaceted approach, incorporating economic opportunities and health education, especially among women, and allowing CHWs to counsel households with conflicts.

## Introduction

Undernutrition among children under five is a significant public health problem leading to increased risks of illness and mortality in children [1]. According to the UNICEF, WHO and World Bank Group's global levels and trends in child malnutrition report of 2021, 194.6 million children suffer from undernutrition, with 149.2 million stunted while 45.4 million are wasted [2]. According to Black et al. [3], undernutrition contributes to 11% of the global burden of disease and can severely inhibit short-, medium-, and long-term child development [4]. In sub–Saharan Africa, approximately 61.4 million children were reported stunted in 2020 [5]. Although several efforts have been made to reduce the prevalence of undernutrition, disparities across regions persist, with Asia and sub-Saharan Africa having the highest prevalence [2, 6]. Furthermore, a 2021 World Bank report suggested that the COVID-19 pandemic may have worsened undernutrition in Asia and sub-Saharan Africa by interrupting food systems and income opportunities [6].

To reduce the burden and prevalence of childhood undernutrition in sub-Saharan Africa, the WHO has adopted strategic plans with a focus on strengthening evidence-based policies and national capacity [7]. Furthermore, the African Union Agenda 2063 continental initiative includes the Comprehensive African Agricultural Development Programme, which seeks to address undernutrition in Africa by increasing food supply and reducing hunger [8]. Additionally, several African countries have launched programs and activities to address undernutrition. In Rwanda, several projects, including *Girinka* [9], USAID *Gikuriro* [10] and Inclusive Nutrition and Early Childhood Development (INECD)-*Gikuriro Kuri Bose* [11], aim to address childhood stunting. However, stunting remains a problem in Rwanda, with a prevalence of 31.4% [12, 13]. This highlights the need to go beyond quantitatively understanding the macro drivers and focusing on family dynamics and cultural beliefs that may influence stunting and its reduction efforts.

Several national-level analyses have explored the determinants of stunting in Rwanda [14, 15]. These studies have provided important insights into the individual, household, and macro drivers of stunting as well as documenting and proposing critical top-down policy directions to enhance stunting reductions. However, there is a paucity of information on the appropriate personal, family, sociocultural, and context-relevant factors. These factors may be bottlenecks in reducing stunting among children under five years, especially in high-risk population areas [12, 14]. This qualitative study explored the lived experiences of women and men to identify issues that are key to influencing childhood nutrition and stunting and explore possible solutions to address the problem. This study was part of the large consortium USAID-funded study "Inclusive Nutrition and Early Childhood Development (INECD)-*Gikuriro Kuri Bose*."

## Methods

### Study setting and population

This study was conducted in five districts of Rwanda: Kicukiro, Ngoma, Burera, Nyabihu, and Nyanza. The prevalence of stunting in these areas ranges from 46% in Nyabihu to 18% in

Kicukiro [13]. Kicukuro is an urban district, whereas Ngoma, Burera, Nyabihu, and Nyanza are rural districts. Farming is the main source of livelihood in all these districts.

## Data collection

A qualitative exploratory approach was used to explore family, community, and sociocultural opinions, beliefs, and key concerns about child nutrition. Data were collected through in-depth individual interviews with key informants (nurses and CHWs) and focus group discussions (FGD) with mothers and fathers of children under the age of five. In Kicukuro, Ngoma, Burera, and Nyabihu, data were collected between October and November 2022, whereas in Nyanza, data were collected in January 2023.

## Sample and sampling

In each district, only facilities which provided nutrition programs from *Gikuriro Kuri Bose*, with nurses and CHWs who met the eligibility criteria, were included in the study. As these facilities and the number of eligible participants were small, participants were conveniently sampled.

**IDI.** Forty (40) in-depth interviews (IDI) were conducted depending on data saturation. The literature shows that, data saturation can be achieved after interviewing 25–30 participant [16] but because of a wider geographic coverage for this project we extended to 40 participants to make sure no new information is left out. In each district, four community health workers from different sectors within the district and four nurses from different healthcare facilities participated in the IDI. Nurses who worked in childhood immunisation programs at health facilities and CHWs who oversaw integrated Community Case Management (iCCM) for childhood illness were interviewed as key informants. Nurses and CHWs with less than six months of working in the study district and those who did not consent to participate were excluded from this study.

**FGD.** Participants of the FGDs included men and women who were above the age of 18, had experience of being a caregiver for a child under the age of 5 years, and resided in the study district for more than 6 months. A total of ten (10) FGDs were held: one for mothers and one for fathers in each of the five districts. Study participants included in the FGD were selected with the help of community health workers. To avoid a selection bias, only one participant from the same household was recruited.

## Data collection tools and data collection

**FDG.** A semi-structured discussion guide (S1 File) was developed and used for the FGDs. FGDs were conducted at health facilities, as they were the more central locations. To avoid disturbances and create an enabling environment for the participants, all FGDs were conducted in the meeting rooms of the health facilities under closed doors to ensure privacy. Each FGD comprised 8 participants. All FGDs were conducted during weekdays between 14:00–16:00 hours for all rural areas and 8:00–12:00 hours for the Kicukiro district to avoid interfering with their farming activities which usually occur in the morning until midday.

Housekeeping rules, details of the study, and rights were provided to all participants before verbal and written consent for participation and recording were obtained.

FGDs were conducted in Kinyarwanda, a nationally spoken language in Rwanda. All FGDs were facilitated by trained research assistants and one researcher (TU) who was fluent in Kinyarwanda. Furthermore, the researcher coordinated the discussions and took notes, in addition to the recordings. This researcher (TU) helped the research assistants collect participants' demographic characteristics, education level, socioeconomic category (Ubudehe category), employment status, number of children under the age of five, and marital status.

Each FGD lasted approximately 60–90 min. At the end of each FGD, participants were reimbursed for the transport costs incurred.

**IDI.** These were also conducted at health facilities which were the most convenient places for the participants, and an ID guide was used to obtain data from them (S1 File). All participants who took part in the study signed the consent forms. The IDI lasted approximately 45 min. The study participants were also reimbursed for transport costs at the end of the IDI.

## Ethical approval and consent to participate

The study was approved by the University of Global Health Equity (UGHE) Institutional Review Board (UGHE-IRB: Ref: UGHE-IRB/2022/034). All participants provided written informed consent to share their anonymised data in the manuscript.

## Data management

All audio recordings were transferred to a password-protected computer and deleted from the audio recorders. The audio recordings were transcribed and translated into English. The transcripts were checked for accuracy and consistency by the lead researchers (GI, MQA, and SM). Observational notes were used to check accuracy and consistency.

## Data analysis

The transcripts were transcribed verbatim and translated into English. Transcripts were read and coded by five researchers (MAQ, SM, GI, NN, and TU) first independently to draft a codebook and then came together to consolidate their codes into a final codebook. This was done manually. Further inductive coding was performed using Dedoose software (version 9.0.86) [17]. Thematic analysis was used to identify important patterns in the data, as described by Braun and Clarke [18]. The team then met to discuss and finalise emerging themes and subthemes. After coding, similar codes were grouped into subthemes and similar subthemes were grouped to themes as shown Fig 1.

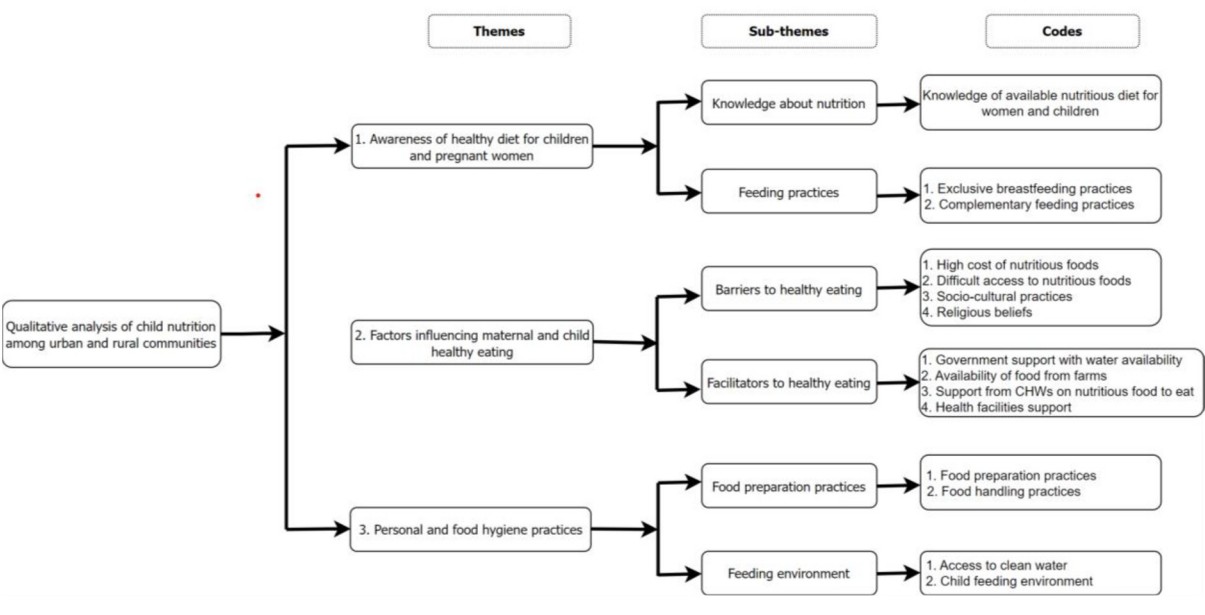

**Fig 1. A coding tree.**

### Scientific rigor

**Credibility.**   Through regular meetings during the data collection for another quantitative project, the research team was able to build rapport with the study participants and this good relationship prior to the interview helped make the participants feel at ease and provide credible responses. Additionally, we collected more data above the data saturation which was key in obtaining sufficient data to answer the research questions. After the completion of audio transcription, four members of the research team reviewed the randomly selected interview scripts and validate the results with the participants' responses captured in the audio recordings. They also conducted coding, sub/themes generation independently and held peer briefing sessions to discuss and agree on the themes/subthemes. To triangulate the responses, researchers include different groups of respondents (caregivers, nurses, and CHWs), and used different data collection methods namely interviews and focus group discussions on the phenomena studied.

**Transferability.**   This was achieved by developing appropriate questions seeking for appropriate responses regarding the caregivers' and key informants' perceptions on maternal and child nutrition Additionally, the inclusion of respondents from a wider geographic area representing the five provinces of Rwanda both urban and rural, provided a wide range of information and recommendations that can be generalized and transferred to other setting with similar characteristics.

**Dependability.**   After the tools were developed, they were reviewed by the nutritionist and Gikuriro Bose project experts to ensure that the questions included in the guide comprehensively captured key areas of maternal and child nutrition. Researchers' peer participation in data collection, analysis and interpretation enhanced dependability of the findings as this reduced researcher personal influence on the study. To ensure that the research was sensitive to sociocultural factor that would affect the outcomes, data was collected from mothers and fathers separately to increase responsiveness of women than when they were with men in the same group. This study was designed in a manner that data was collection from one caregiver per household to expand the scope of responses from various households.

**Conformability.**   Conformability was achieved by making sure that the findings reflected the participant's voice and conditions of the inquiry, not the biases, assumption or the perspectives of the researchers. Conformability of data is critical to ensure the objectivity in the study. Through reflexivity the researchers regularly met to continually reflect, appraise and critique their roles, biases and assumptions during data collection, data analysis and interpretation furthermore kept a neutral position from interfering with the research outcomes and the responses. The researchers used a reflective coding book to capture respondents' personal perceptions about the phenomenon being studied such that they did not influence the interpretation of the results. During data analysis, audit follow up was done by handing over the raw data and coding book to the Primary Investigators of this study to cross check and to confirm trends in the data.

## Results

### Participants demographics

Ten FGDs were conducted, two per district, with eight participants in each FGD, for a combined total of 80 participants. Forty (40) IDI were conducted, consisting of 20 nurses and 20 community health workers (CHWs), with four individuals from each profession per district. The gender distribution among the participants was 45.8% male and 54.2% female. Participants' ages ranged from 19 to 62 years, with the majority (63.8%) having attained primary

school education and belonging to the *Ubudehe* socioeconomic category 2. Among the FGD participants, the majority (53.8%) were farmers and 20% were unemployed. Furthermore, 40% had one child under five years old, and 82.5% were married. A coding tree was generated using an inductive approach to identify the codes, subthemes, and themes that emerged from the data, as shown in Fig 1.

The following themes emerged from the data.

Theme 1. Awareness of a healthy diet for pregnant women, infants, and children.

- Subtheme 1; Knowledge about nutrition

- Subtheme 2; Feeding practices

Theme 2. Factors influencing healthy eating among pregnant women, infants, and children.

- Subtheme 1. Barriers to healthy eating among pregnant women and children

- Subtheme 2. Facilitators to healthy eating among pregnant women and children

Theme 3. Personal and food hygiene is crucial while handling, preparing, and eating food

- Subtheme 1; Food preparation practice

- Subtheme 2; Feeding environment

**Theme 1**. **Awareness of a healthy diet for pregnant women, infants, and children.**
Subtheme 1; Knowledge about nutrition

Participants demonstrated knowledge of the types of foods that constitute a nutritious diet for both pregnant women and children. Among the foods commonly cited as nutritious for pregnant women are those that provide energy, build the body, and offer protection, such as fruits, vegetables, and legumes.

*"Pregnant woman is supposed to take energetic food including sweet potatoes, cassava and rice, the body building food include beans, animal products, and protective food include generally fruits and vegetables"*

*(Female FGD, KicukiroDistrictt).*

*"A pregnant woman is recommended to take green vegetables, meat, rice, fruits mostly those that increase blood in the body so that she can keep being better."*

*(Male FGD Nyabihu district)*

The participants understood the importance of a healthy and nutritious diet for pregnant women, stating that it promotes the health of both the mother and child, as well as supports the growth of the unborn child.

*"All those foods can help the mother to be healthy and the baby in the womb to grow well and receive the required nutrients…."*

*(Female FGD, Kicukiro district)*

*"… these foods are better for a pregnant woman since they contain all nutrients needed by the pregnant woman and the baby in the womb."*

*(Male FGD Burera district)*

Subtheme 2; Feeding practices

Most participants seemed to understand the importance and delivery of exclusive breast-feeding. They could clearly state that a child should be exclusively breastfed for the first six months without any additional food, and then introduce complementary food after this time. They also stated that a child can be breastfed as often as needed during exclusive breastfeeding, and breastfeeding can continue for up to two years before weaning.

*"From birth to the age of six months, the baby should be given nothing else except breast milk only after that complimentary food can be started."*

*(Female FGD Nyanza district)*

The initiation of complementary foods was determined by child growth and maternal health status according to the participants. They understood that breastmilk does not provide enough nutrition for a child older than six months and thus introduced complementary foods such as soft porridge and a mixture of various light foods, as a child's stomach is not strong enough to digest heavy foods.

*"The child should be given milk and then the SOSOMA porridge, this is because a child's stomach and his digestive system are not able to process solid food like that of an adult."*

*(Male FGD Burera district)*

*"As the child grows, they are introduced to other foods as the mother continues to give breast milk until he turns two years old."*

*(Female FGD Nyabihu district)*

Others stated that sometimes the child can be introduced to complementary foods earlier than six months, when the mother cannot continue breastfeeding the child.

*"When the mother has died or is suffering from a certain disease with which she cannot breastfeed her baby in that case you can give your baby other foods such as milk earlier than even 6 months."*

*(CHW Nyanza district)*

Theme 2. Factors influencing the health eating among pregnant women, and infants and children.

**Subtheme 1.** Barriers to healthy eating among pregnant women and children

While the participants generally had a good knowledge of nutritious foods, they indicated difficulties in accessing these foods for financial reasons. Milk, eggs, and chicken meat were considered foods for wealthy people and were rarely consumed by the participants.

*"If parents have money, they will buy milk for their children. However, if you do not have money,! I would not say that they drink milk, or where would they get it? Milk is for rich people; they are the ones who drink milk."*

*(CHW Kicukiro district)*

*'I give my child milk and sometimes when God has been good which is rare, I afford the meat and I give them the soup."*

*(Female FGD Kicukiro district)*

Despite knowing that children should be fed at least three to five times a day, many participants were unable to do so because they were on the farm all day, making it difficult to access nutritious foods and could only feed their children once a day, which they recognised could lead to stunted growth.

*". . .. We should feed a child who is six to eight months old three times a day, but in reality, when we are in the field, cultivating, a child can only eat once a day. This is because work hours are long. When we started feeding a child once a day, he started to stunt. Therefore, we ignore it, yet we know it."*

*(Female FGD Burera district).*

In addition, nutritious complementary foods, such as milk and eggs, are inaccessible for children to consume, even among households with cow(s) producing milk and chickens laying eggs. All were sold for income without reserving any children.

*"Some people have a milking cow and deliver all the milk for sale; their children do not drink milk. And who has chicken can't eat an egg, because all are taken to the market for sale."*

*(Male FGD Burera district)*

Participants reflected on the idea that some religious beliefs do not allow women and children to consume foods, such as meat and milk, regardless of how nutritious these foods are. Thus, nurses mentioned that pregnant women faced with these taboos sometimes suffer from poor nutritional health outcomes such as anaemia.

*"Adventists, I heard, do not eat meat, do not drink milk, I have heard that their children are not given meat and milk."*

*(CHW Kicukiro district)*

*"A pregnant woman can face the issue of anaemia because of eating an unbalanced diet."*

*(Nurse Nyanza district)*

Most participants emphasised that the use of alcohol and drugs during pregnancy can result in adverse health effects on them and the developing foetus, thus avoiding consumption of alcohol and any kind of drug, including medication, unless it is prescribed by a doctor. Some individuals provide alcohol to their children with the belief that they can prevent intestinal worms.

*". . . it is taboo for a pregnant woman and a child to take alcohol. If a pregnant woman consumes alcohol more frequently during pregnancy, she may give birth to a mentally abnormal baby."*

*(Male FGD Ngoma district)*

*"A pregnant woman is not allowed to take any type of medicine without the doctors' recommendation, not even traditional herbs because that might cause her to miscarriage or even risk her life to death."*

*(Nurse Nyanza district)*

*"No child is allowed to take alcohol. We always warn parents not to give their children alcohol because they say that they serve children alcohol for preventing intestinal worms."*

*(CHW Burera district)*

Most women stated that even if they prioritise their children and their nutrition during pregnancy, they need to ensure that their husbands always have food. If their husbands were not around them, they would reserve a portion of their husbands' food.

*"I put aside the portion of food for my husband so that I can serve the kids and feed them without fear of spoiling the husband's food."*

*(Female FGD Ngoma district)*

Participants, mostly women, mentioned that a bigger piece of meat was reserved for the husband due to cultural respect and believed that meat was not good for children. However, the male participants expressed that they should receive a larger portion of meat because they worked hard to afford it.

*"When we have served meat, my husband comes first, he must be given respect so he takes the bigger part, then my kids can get some. Also, feeding a little child meat is not good, what is important is the soup which contains the main nutrients."*

*(Female FGD Kicukiro district)*

*"For us to afford the chicken, of course, he must have worked for it. After all, he (a man) didn't eat out but chose to share it with his family then a man should get the biggest piece as he is the one who worked hard to get the chicken, but also children should get their share."*

*(Male FGD Nyabihu district)*

However, some participants held different views on this topic. In these households, it was not considered important to consume a larger portion of meat. Each family member served and ate the desired amount.

*"In our home, we are equal, we do not go into bigger slices goes to who or what, if I have cooked meat, we eat together whoever picks the thigh eats it, but the husband cannot say that he only gets the bigger portion or me serving him that. You eat what you pick when serving yourself."*

*(Female FGD Kicukiro district).*

Poverty often causes conflicts in food distribution and negatively affects children's feeding and growth. Children living in households with insufficient food are often unhappy and experience poor growth. This also causes a lack of happiness in households. When there is a shortage of food in the household, it often leads to conflicts and disagreements. Children in families with conflicts lack peace, which affects their appetite, and ultimately results in poor growth.

*"If I eat a piece of meat that was preferred by my husband and he does not find it, I am not lying, he will make me pack my things and go back to my family with no discussion, conflicts are there."*

*(Female FGD Ngoma district)*

*"We are poor, nothing goes well, we are not happy with each other in the house, nobody wants the children to grow unhealthy but the means to afford the foods is the problem."*

*(Female FGD Kicukiro district)*

*"If my husband goes to work, instead of bringing money at home, he goes straight to the bar and uses all the money and nothing to eat in the house there will be a fight in the house. Do you think that a child who does not have food can be happy? That child can also end up not growing'*

*(CHW Kicukiro district).*

*"So, a family with conflicts has no peace, and their children get stunted easily because the child will not eat, that child will not be happy, and it will be difficult to finish food and then not grow"*

*(Male FGD Ngoma district).*

Subtheme 2. Facilitators to healthy eating among pregnant women and children
The respondents appreciated the efforts made by government of Rwanda in supporting the communities with nutritious complementary porridge locally known as "*shisha kibondo*", providing financial incentives to improve maternal and child nutrition, training community health workers and community members on maternal and child nutrition, influencing the nutritional support from early childhood development services and supporting the provision of safe drinking water.

*". . .it has been easy today the government has brought for us 'shisha kibondo' porridge and some money to complement and make it easy to afford fruits and other needs.*

*(Female FGD Nyabihu district)*

*"The government has done a good thing by bringing about the policy that encourages us to take care of our pregnant wives and the children, even though it's not hundred percent as they trained community health workers and then they trained people on nutrition . . ."*

*(Male FGD Nyabihu District)*

They further stated that in recent years, early childhood development services have made nutritious foods available for children, such as milk. Through government support, some communities were able to access drinking water from taps, and filters were made available to ensure water portability.

*"Our children have recently been able to access milk in newly introduced ECDs. . .*

*(Male FGD Ngoma District)*

*"They (government) installed drinking water taps . . .and provided us with water filters to make sure we are able to get clean drinking water"*

*(Male FGD Nyabihu district)*

**Theme 3**. **Personal and food hygiene is crucial while handling, preparing, and eating food**
Subtheme 1; Food preparation practice

Participants were aware that poor hygiene during food preparation could introduce pathogens into food, causing foodborne illness. They were conscious of food hygiene practices, such as washing food, using washed utensils, and personal hygiene, such as hand washing, especially before feeding their children.

*". . . before preparing that food, it is okay to wash hands and wash food too with clean water, cooking food in clean utensils even serve the food on clean plates."*

*(Female FGD Ngoma district)*

*"Prepare the food with cleanliness and feed it using clean materials like preparing a washed plate, a clean spoon, and a clean cup that is dry so that there no related disease-causing micro-organisms. "*

*(Female FGD Nyabihu District)*

However, their work environment does not allow them to practice hygiene before feeding or breastfeeding. When they work on a farm, their children are left to feed themselves, mixing their food with soil and dirt.

*"Without hygiene when preparing or serving foods will result to bad bacteria getting in food and making children and others have their stomach-ache."*

*(Male FGD Nyanza district).*

*"When we are in the field cultivating, we do not find water to clean our hands. You give your child a sweet potato quickly because there is no water, and you continue cultivating it. In time for breastfeeding, sometimes you take a child and breastfeed him without washing your hand. You simply rub your hands with one another to reduce dust. For real, we do not respect hygiene well."*

*(Female FGD Burera district)*

*"We carry porridge and food in the small jerry can. If he cries, you place a dish full of food in front of him and feed him. After a certain time, if you come back to him, you realise that he places the soil in his food and eats a mixture of food and soil. No hygiene of food indeed."*

*(Female FGD Burera district)*

Subtheme 2; Feeding environment

Some participants suggested bringing a spoon to feed their children while working on the farm when water access was limited. However, others expressed concerns that bringing a spoon could result in them losing their jobs, particularly if they were working on someone else's land.

*"If you remember to bring food and vegetables to the farm, you can bring a spoon too. This spoon helps you to feed a child in case you are dirty and there is no water to wash hands."*

*(Female FGD Burera district)*

*"Some of us cultivate for others, there is no way you can go to cultivate in the field with a spoon to feed your child, yet it is not your field. You cannot do this because your boss can fire you and [you would] lose a job."*

*(Female FGD Burera district)*

The participants were generally aware of the importance of safe clean water. They noted that some of the water they drink may be exposed to disease-causing organisms. Some were able to access potable water in their community, while others obtained their water from unsafe sources and needed to be treated through boiling.

*". . .we are still facing the problem of safe water because we continue to use and drink swampy water, which may be a source of intestinal parasites."*

*(Male FGD Burera district)*

*"Now my family has access to potable water because there is a tap, and community health workers always encourage us to boil water because it becomes safe for drinking. "*

*(Female FGD Nyabihu district)*

## Discussion

Nutritional knowledge is one of the pillars of nutrition literacy [19] and a lack of knowledge necessitates interventions such as community nutritional education. Our study participants were knowledgeable about the need to provide nutritious food to pregnant women and their children to promote good health. The respondents correctly cited locally available foods that constitute a healthy diet essential for pregnant women and unborn babies' well-being, including energy-giving, body-building, and body-protecting foods, with a list of correct examples for each. They also stated that, the increased energy and food needs of the mother during pregnancy and the fetus, make eating healthy even more important [20]. They also understood and practiced exclusive breastfeeding and complementary feeding.

A high level of knowledge of child feeding practices may be attributed to intensive community nutrition-based education [21, 22]. In Rwanda, CHWs provide continuous nutritional counselling and behavioural change communication which is crucial for improving breastfeeding and young child feeding practices at the household level [23]. The engagement of CHWs in empowering communities with nutrition-based knowledge is a successful initiative of the Government of Rwanda and is worthy of replication in areas with poor knowledge of maternal and child nutrition.

High knowledge of healthy foods does not directly translate into healthy food practices. Many of our participants had difficulties in affording and accessing healthy and diverse diets due to poverty, an outcome that has been reported in other studies [22, 24]. Milk, meat, and eggs were only occasionally consumed, even for households that reared animals, as animal products were mainly sold, which is consistent with previous observations [25]. Recently, the Government of Rwanda and the United Nations established a joint program that focuses on improving household food security and dietary diversity through small livestock rearing, kitchen garden construction, income-generating activities, and savings [23]. UNICEF, in collaboration with the Government of Rwanda, has an ongoing clean water supply program for rural households [26]. Additionally, non-governmental organizations such as Catholic Relief Services (CRS) have, over the years, supported communities in Rwanda with services that improve maternal, infant and child nutrition and development through the Inclusive Nutrition and Early Childhood Development project known locally as "*Gikuriro Kuri Bose*" [27]. These programs are envisaged to contribute to a diversified diet which is essential for the reduction of undernutrition, including for stunting children (Khamis et al., 2019). Without sufficient

variety in their diets, children cannot obtain the adequate nutrients needed to grow effectively [26].

Evidence has shown that lower meat consumption among pregnant women and children is associated with negative nutritional and growth outcomes [28]. The participants reported that some religious beliefs and sociocultural practices prohibited women and children from consuming certain foods, and similar restrictions were also reported in other settings [29]. Such restrictions on meat consumption can affect the health of pregnant women and their children. There is a need to engage religious and community leaders on the importance of consuming diverse foods for pregnant women and children. We also suggest that future studies should assess the extent to which religious beliefs affect the nutritional status of pregnant women and their children.

Food distribution and quantity allocation have been discussed in other studies [30]. In this study, some participants reported that they would serve their husbands first, while others served the children first, and some even kept food aside for their husbands before distributing it to their children and themselves. While the serving order may be practiced differently, it seemed universal that all of them serve a bigger portion of the meat to the husbands. Similar to many other studies, food distribution patterns seem to favour adults, especially fathers, which may lead to poor health outcomes for other members of the household, such as children [31]. Community nutrition-based education should incorporate the concept of equity into household food distribution.

Respondents also indicated that food preparation and feeding hygiene practices must be performed with clean water, but most of them relied on water from unsafe sources. The availability of reliable clean water, including boiled water, is key to child health improvement [32]. To enhance access to clean drinking water in rural communities, the government of Rwanda envisaged providing universal water services through its Water Supply Policy. However, there have been challenges in scaling up this initiative in scattered rural settlements [33]. It is crucial to ensure access to clean water and proper sanitation facilities, as these are fundamental for preventing waterborne diseases and supporting overall health and nutrition. The feasibility of incorporating the private sector and its cooperating partners to support this government initiative should be explored.

Conflict/intimate partner violence (IPV) was related to poor feeding among women and children [34]. Some respondents stated that conflicts often result in a lack of peace among children, leading to a loss of appetite and ultimately resulting in poor growth [34, 35]. The current study suggests that community healthcare workers should be involved in counselling families, especially those with frequent domestic violence, to increase the risk of poor health and growth outcomes among women and children. Families should be made aware of the impact of IPV, including its permanent health consequences for children.

Improvements in access to a healthy diet and nutritious food will be an important driver in reducing stunting in children. To address barriers such as household personal and food hygiene, basic-level capacity-building needs to be enhanced during prenatal and antenatal care visits. The results of our study suggest the need for government and cooperating partners' stunting reduction program-related activities to be complemented with education and promotion among mothers with an emphasis on parenting and nutrition. Although Rwanda has initiated systematic training of healthcare providers [36], it is necessary to extend this capacity-building training in nutrition to other community members, especially women with children under five.

The main limitation of the study is that it was conducted in only five of the ten districts where the *Gikuriro Kuri Bose-Inclusive* Nutrition and Early Childhood Development (INECD) is currently taking place and where stunting is a problem. Including other districts where

stunting has greatly reduced below the national average may have offered us insights into some of the lessons and activities that have worked and potentially need to be upscaled.

Despite this limitation, the current study provides important information and builds on the suggestions of other previous studies on the drivers of stunting in Rwanda. This study also revealed the insights and lived experiences of the rural and urban populations and those of service providers on factors influencing child and maternal nutrition. The findings of this study will be vital in guiding policymakers in designing and planning nutrition programs to further reduce child stunting in Rwanda.

## Conclusions

This qualitative study revealed that respondents had high knowledge of food and feeding practices. Financial constraints in the study communities were the main barriers to providing healthy and diverse diets to pregnant women and children. The availability of safe clean water was also a challenge for some participants in implementing personal and food hygiene practices. A multidimensional and multisectorial approach is required to enhance diversification in food production, women's economic empowerment, and scaling up clean water availability and access in rural communities.

## Supporting information

**S1 File. Focus group discussion and in-depth interview guide.**
(DOCX)

**S1 Dataset. PLOS dataset.**
(DOCX)

## Author Contributions

**Conceptualization:** Maria Qambayot Albin, Abebe Bekele, Rex Wong, Chester Kalinda.

**Data curation:** Shuko Musemangezhi, Edith Nachizya Namukanga, Theogene Uwizeyimana, Chester Kalinda.

**Formal analysis:** Maria Qambayot Albin, Gloria Igihozo, Shuko Musemangezhi, Edith Nachizya Namukanga.

**Funding acquisition:** Gebremariam Alemayehu, Abebe Bekele, Rex Wong.

**Investigation:** Maria Qambayot Albin, Theogene Uwizeyimana.

**Methodology:** Maria Qambayot Albin, Theogene Uwizeyimana, Rex Wong, Chester Kalinda.

**Validation:** Shuko Musemangezhi.

**Visualization:** Shuko Musemangezhi.

**Writing – original draft:** Maria Qambayot Albin, Gloria Igihozo, Edith Nachizya Namukanga, Gebremariam Alemayehu.

**Writing – review & editing:** Rex Wong, Chester Kalinda.

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
