## [Decision Letter · Decision Letter 0]

9 Jan 2024

PONE-D-23-33374“ When we have served meat, my husband comes first”: A qualitative analysis of child nutrition among urban and rural communities of Rwanda.PLOS ONE

Dear Dr. KALINDA,

Thank you for submitting your manuscript to PLOS ONE. After careful consideration, we feel that it has merit but does not fully meet PLOS ONE’s publication criteria as it currently stands. Therefore, we invite you to submit a revised version of the manuscript that addresses the points raised during the review process.

We look forward to receiving your revised manuscript.

Kind regards,

Fatch Welcome Kalembo, Ph.D

Academic Editor

PLOS ONE

Journal Requirements:

"USAID Rwanda through the Gikuriro Kuri Bose - Inclusive Nutrition and Early Childhood Development (INECD)"

"The authors would like to acknowledge the financial support of USAID Rwanda through the Gikuriro Kuri Bose - Inclusive Nutrition and Early Childhood Development (INECD) Program given to CRS and University of Global Health Equity (UGHE)."

"USAID Rwanda through the Gikuriro Kuri Bose - Inclusive Nutrition and Early Childhood Development (INECD)"

"None"

6. In the online submission form, you indicated that [Data can be availed upon request and by following the data sharing guidelines of the University of Global Health Equity (UGHE), Catholic Relief Service (CRS) Rwanda office and the National Child Development Agency (NCDA) of Rwanda own the data.]. 

7. Your ethics statement should only appear in the Methods section of your manuscript. If your ethics statement is written in any section besides the Methods, please move it to the Methods section and delete it from any other section. Please ensure that your ethics statement is included in your manuscript, as the ethics statement entered into the online submission form will not be published alongside your manuscript. 

Additional Editor Comments:

Abstract

• The aim of the study is not clear in the abstract

• Please provide the number of in-depth interviews that were conducted with

Introduction

• Please specify the context (global or regional) of the stats in the second sentence of paragraph 1.

• The first paragraph could also be made stronger by including child undernutrition/malnutrition statistics for sub-Saharan Africa and Rwanda. This could help the reader to appreciate the extent of the problem

• The first paragraph lasts but one sentence ‘Although several efforts have been put in to reduce the prevalence of undernutrition, disparities across regions persist with Asia and sub-Saharan Africa having the highest prevalence’ This sentence needs editing

Methodology

• How were the sample sizes for the focus group and in-depth interviews determined?

• The process of data analysis needs further details. Could you please explain how you used Braun and Clarke (2018) to analyse the data

• What measures did you use to ensure the trustworthiness of the data?

Findings

• Please explain how you developed the five themes from the initial codes (a coding tree could help to illustrate this).

• Please consider revising and rewording your themes. They need to be clear and concise. For example, themes 2, 3 and 5 could come under one theme ‘Barriers to accessing nutritious food’ or ‘Perceived barriers to healthy eating ‘- with subthemes.

Discussion

• Providing clear recommendations for practice, research and policy could make this section stronger. I am aware that some implications for practice have been provided however these are scanty and brief.

Reviewers' comments:

Reviewer's Responses to Questions

**Comments to the Author**

1. Is the manuscript technically sound, and do the data support the conclusions?

Reviewer #1: Yes

2. Has the statistical analysis been performed appropriately and rigorously? 

Reviewer #1: N/A

3. Have the authors made all data underlying the findings in their manuscript fully available?

Reviewer #1: No

4. Is the manuscript presented in an intelligible fashion and written in standard English?

Reviewer #1: Yes

5. Review Comments to the Author

Reviewer #1: Title: “When we have served meat, my husband comes first”: A qualitative analysis of child nutrition among urban and rural communities of Rwanda.

Authors: Albin Maria, Igihozo Gloria, Musemangezhi Shuko, Namukanga Edith, Uwizeyimana Theogene, Gebremariam Alemayehu, Abebe Bekele, Wong Rex & Kalinda Chester.

Dear Authors,

Thank you for giving me the opportunity to review your manuscript entitled “When we have served meat, my husband comes first”: A qualitative analysis of child nutrition among urban and rural communities of Rwanda”. The author has executed and reported an interesting research assessing the strategies and methods used to investigate the prevalence of stunting/undernutrition among children in Rwanda. Overall, the information presented in the manuscript represents valuable information. However, the manuscript would benefit from a well-detailed description of the methodology and findings to improve understanding and flow. I feel this unique dataset has not been utilized to its full extent. Below I have provided numerous remarks on the text. Given these shortcomings the manuscript requires major revision.

Abstract:

Do you refer to childhood or stunting or undernutrition as an issue of public concern

Spell out the acronym FGD: Focus Group Discussions to allow the reader to understand

You mean data were analyzed thematically not collected. Can you specify please the methodology used to analyse the data

What does CHW stands for?

Not sure if the title on husband fits as we are talking about children in the manuscript

Introduction:

Thank you for your succinct introduction. No comments. I like the idea of stating why you wanted to go to investigate the issue qualitatively as opposed to quantitatively.

Just one comment. Have you ever thought of including social cognitive theory into your introduction? This is a very interesting theory that shows there are a lot of factors that could impact person health.

Methods:

Very nice to distinguish between urban and rural areas.

Explain why you chose to do FGR with one group and in-depth interviews with the others.

What are your stakeholders? Having a clear description is better? As I am currently confused. Have you ever thought of having a table?

Have you used the same guide for both the in-depth interviews and the FGD?

Where is closed room? Research facility or in the farm?

Who conducted the interviews/FGD? The main researcher or assistant? Also, do you mean by observation notes of the farm or the interview notes in addition to recording?

Who have done the translation? A company service or one of the researchers?

Very beautiful Methods. Just make it clearer. You have all the information, but the flow is not there. Thank you very much

Very beautiful using Braun and Clark. I have used it myself. Very well done. Did you perform your analysis both manually and using the software.? More information on the methodology used for both would be appreciated.

Results:

Having a table would facilitate the understanding who are your key stakeholders. Pregnant women, farmers, etc.

Very nice integration of quotes into the main text. Well-done.

Have you thought of including if this quote is related to urban or rural area? Their views might be different. Just a thought here.

You can also reduce some of the wording in the actual quote by putting only three dots, not four or five.

Discussion:

You can say reported or stated instead of understood

You can start your discussion with a paragraph that state/summarize the main findings of your study and then you can go to back your main findings with the literature

Liked the ideas of using respondents, interviewees, etc.

Very nice discussion. Just make it flow more coherently. You can also put two sections in there. Implications for research and implications for practice. I can see that you have incorporated something there about “the results from our study suggest that need for government and cooperating ……” You can put this into a new subtitle into your discussion part of the paper just to have your part easily laid-out “friendly” to flow logically

Make sure you state more strengths. You have lots there. Such as using qualitative methodology to better understand the issue

Conclusions:

Very beautiful section. No comments at all.

References:

Very good. No comments at all.

6. PLOS authors have the option to publish the peer review history of their article (what does this mean?). If published, this will include your full peer review and any attached files.

Reviewer #1: No

---

## [Author Response · Author response to Decision Letter 0]

9 Feb 2024

We acknowledge receipt of the reviews’ comments for the manuscript that was submitted to your journal for publication. Kindly find below a point-by-point rebuttal of the comments that were addressed

---

## [Decision Letter · Decision Letter 1]

6 May 2024

PONE-D-23-33374R1“When we have served meat, my husband comes first”: A qualitative analysis of child nutrition among urban and rural communities of Rwanda.PLOS ONE

Dear Dr. KALINDA,

Thank you for submitting your manuscript to PLOS ONE. After careful consideration, we feel that it has merit but does not fully meet PLOS ONE’s publication criteria as it currently stands. Therefore, we invite you to submit a revised version of the manuscript that addresses the points raised during the review process.

We appreciate your efforts in addressing the majority of the reviewers' feedback. While there has been an improvement in the manuscript's quality, further refinement is necessary before it meets the standards for publication. Apart from incorporating the reviewer's suggestions, we kindly ask you to also consider the following:

The manuscript is thoroughly edited. There are still some grammatical and punctuation errors in the manuscript. A few examples of these include:Lines 31-32 ‘Transcripts were coded inductively analysed thematically using Dedoose (version 9.0.86)Line 51-52 ‘In sub–Saharan Africa, approximately 61.4 million children were stunted by 2020’Lines 66-69 ‘However, stunting remains a problem in Rwanda, with a prevalence of 31.4% [12, 13] further highlighting the need to go beyond quantitatively understanding the macro drivers and focusing on family dynamics and cultural beliefs that may influence stunting and its reduction efforts.’ This is a long sentence that needs editingLines 94-95 ‘In each district, only facilities which provided nutrition programs from Gikuriro Kuri Bose with 95 nurses and CHWs who met the eligibility criteria were included.’ This needs rewordingMethodology: Page 4 line 96 ‘As these facilities and the 96 number of eligible participants were small, no sampling was performed.’ Sampling can still be conducted with a small sample size. You are required to provide information about the sampling technique you used to recruit participants to the study. For example, did you use purposive or convenience sampling?Data analysis: Although there is a mention that data were analysed through Braun and Clark- thematic analysis, there is not sufficient information to enable the leader to understand how data analysis was done. Please provide more information about how data analysis was done thematically.Results: For the coding tree (Fig 1) The last theme has subthemes. What are the subthemes for the first and second themes? Please consider providing subthemes for the first and second themes. For example, the subthemes for the first theme can be 1) Feeding practices and 2 Knowledge of available nutritious diets.The findings section of the abstract has five themes while the main text of the paper has three themes. Please correct this so that the themes in the abstract and the main text are the same. ==============================

We look forward to receiving your revised manuscript.

Kind regards,

Fatch Welcome Kalembo, Ph.D

Academic Editor

PLOS ONE

Reviewers' comments:

Reviewer's Responses to Questions

**Comments to the Author**

1. If the authors have adequately addressed your comments raised in a previous round of review and you feel that this manuscript is now acceptable for publication, you may indicate that here to bypass the “Comments to the Author” section, enter your conflict of interest statement in the “Confidential to Editor” section, and submit your "Accept" recommendation.

Reviewer #2: (No Response)

2. Is the manuscript technically sound, and do the data support the conclusions?

Reviewer #2: (No Response)

3. Has the statistical analysis been performed appropriately and rigorously? 

Reviewer #2: (No Response)

4. Have the authors made all data underlying the findings in their manuscript fully available?

Reviewer #2: (No Response)

5. Is the manuscript presented in an intelligible fashion and written in standard English?

Reviewer #2: (No Response)

6. Review Comments to the Author

Reviewer #2: The following comments:

1.sampling: please indicate in detail characterstics of mothers and fathers of the <5 children-in terms of diveresity such as age, family size, number of children they have, educational status, monogamous or polygamous, nutritional status of the children. How diverse are the FGDs?. Please also indicate to the extent to which the settings or regions included in the study are diverse in terms of factors causing potential nutritional status such as food production, cultural variation in relation to food and nutrition (environmental triangulation). Sample size: clearly indicate about the sample size, how it was fixed? Was idea saturation reached? How was the data triangulation assured during the sampling? What specific sampling technique(s) you employed?

2. Data collection methods: what is the difference between IDIs (in-depth interview) and KIIs (key informant interview). Are the nurses and health workers KII or IDIs? Have you included IDIs from parents of stunted children? How they see the problem? What do they think the factors of the problem? Please clearly locate in the findings about this,if any.

3. What is the name of the qualitative study design you employed? Grounded theory looks to fit more with the research question. Ig so, write the specific design and how did you really implemented the design during the study. For example, thoeritical sampling, data reducation approach.

4. Rigor: please give a separate section for rigor in the.methods. mention how clearly you executed credibility, dependability, transferability, and confirmability. Specifically about: saturation, triangulation, reflexivity, interpretation bias, subjective neutrality, peer debriefying, audit trial, et in the rigor section. You may refer to sampling, tool, data collection, data analysis as a point of refefernce to explain those. Currently much is not said about these points.

7. PLOS authors have the option to publish the peer review history of their article (what does this mean?). If published, this will include your full peer review and any attached files.

Reviewer #2: No

---

## [Author Response · Author response to Decision Letter 1]

3 Jun 2024

Kindly attached an edited version of the submission entitled ““When we have served meat, my husband comes first, must be given respect so he takes the bigger part”: A qualitative analysis of child nutrition among urban and rural communities of Rwanda.” being submitted in consideration for review and publication in PLoS One. We have addressed all comments that were raised by the reviewers.

---

## [Editor Report · Decision Letter 2]

18 Jun 2024

“ When we have served meat, my husband comes first”: A qualitative analysis of child nutrition among urban and rural communities of Rwanda.

PONE-D-23-33374R2

Dear Dr. KALINDA,

We’re pleased to inform you that your manuscript has been judged scientifically suitable for publication and will be formally accepted for publication once it meets all outstanding technical requirements.

Kind regards,

Fatch Welcome Kalembo, Ph.D

Academic Editor

PLOS ONE

---

## [Editor Report · Acceptance letter]

5 Jul 2024

PONE-D-23-33374R2 

PLOS ONE

Dear Dr. Kalinda, 

I'm pleased to inform you that your manuscript has been deemed suitable for publication in PLOS ONE. Congratulations! Your manuscript is now being handed over to our production team.

Kind regards, 

on behalf of

Dr. Fatch Welcome Kalembo 

Academic Editor

PLOS ONE